# Metabolomic Profiling of Antioxidant Compounds in Five *Vachellia* Species

**DOI:** 10.3390/molecules26206214

**Published:** 2021-10-14

**Authors:** Garland Kgosi More, Stephen Meddows-Taylor, Gerhard Prinsloo

**Affiliations:** 1College of Agriculture and Environmental Sciences Laboratories, University of South Africa, Florida, Johannesburg 1710, South Africa; 2Department of Life and Consumer Sciences, College of Agriculture and Environmental Sciences, University of South Africa, Florida, Johannesburg 1710, South Africa; mtayls@unisa.ac.za; 3Department of Agriculture and Animal Health, College of Agriculture and Environmental Sciences, University of South Africa, Florida, Johannesburg 1710, South Africa; prinsg@unisa.ac.za

**Keywords:** natural products, *Vachellia*, antioxidant, metabolomics, ^1^H NMR, UHPLC-qTOF-MS

## Abstract

The genus *Vachellia*, previously known as Acacia, belongs to the family Fabaceae, subfamily Leguminosae, which are flowering plants, commonly known as thorn trees. They are traditionally used medicinally in various countries including South Africa for the treatment of ailments such as fever, sore throat, Tuberculosis, convulsions and as sedatives. The aim of this study was to determine biochemical variations in five *Vachellia* species and correlate their metabolite profiles to antioxidant activity using a chemometric approach. The antioxidant activity of five *Vachellia* aqueous-methanolic extracts were analyzed using three methods: 2,2-di-phenyl-1-picrylhydrazyl (DPPH) radical scavenging assay, 2,2′-azino-bis(3-ethylbenzothiazoline-6-sulfonic acid (ABTS^+^) analysis and the ferric reducing antioxidant power (FRAP) assay by means of serial dilution and bioautography with the thin-layer chromatography (TLC) method. Amongst the *Vachellia* extracts tested, *V. karroo*, *V. kosiensis* and *V. xanthophloea* demonstrated the highest DPPH, ABTS^+^ and FRAP inhibitory activity. The antioxidant activities of DPPH were higher than those obtained by ABTS^+^, although these values varied among the *Vachellia* species. Proton nuclear magnetic resonance (^1^H NMR), coupled with multivariate statistical modeling tools such as principal component analysis (PCA) and orthogonal partial least squares discriminant analysis (OPLS-DA), were performed to profile metabolites responsible for the observed activity. The OPLS-DA categorized the five *Vachellia* species, separating them into two groups, with *V. karroo*, *V. kosiensis* and *V. xanthophloea* demonstrating significantly higher radical scavenging activity than *V. tortilis* and *V. sieberiana,* which clustered together to form another group with lower radical scavenging activity. Annotation of metabolites was carried out using the ultra-high-performance liquid chromatography–quadrupole time-of-flight mass spectrometry (UHPLC-qTOF-MS), and it tentatively identified 23 metabolites of significance, including epigallocatechin (*m/z* = 305.0659), methyl gallate (*m/z* = 183.0294) and quercetin (*m/z* = 301.0358), amongst others. These results elucidated the metabolites that separated the *Vachellia* species from each other and demonstrated their possible free radical scavenging activities.

## 1. Introduction

The etiology of some diseases is due to the inability of a biological system to detoxify oxidative free radicals, leading to the accumulation and imbalances of ROS levels. Diseases such as atherosclerosis, neurodegenerative diseases, rheumatoid arthritis, age-related degeneration and cancer are directly or indirectly initiated by accumulation of ROS [1]. Radicals such as peroxide, hydroxyl and peroxyl radicals that are released by cells often result in degradation of proteins and DNA that facilitates these diseases. Oxidative stress and antioxidant defense mechanisms play a key role during various illnesses instigated by microorganisms including bacteria, viruses and neurodegenerative diseases such as gonorrhea, diarrhea, human immune deficiency virus (HIV), herpes simplex virus (HSV), cancer, Alzheimer’s disease (AD), Parkinson’s disease (PD) and amyotrophic lateral sclerosis (ALS) [2]. Despite the currently available treatments for these ailments, the search for new therapeutic drugs should continue considering that synthetic medicines have demonstrated limited success whilst displaying many adverse side effects. Research into the efficacy of medicinal plants and their constituents to treat neurodegenerative diseases is therefore warranted. Technologies such as metabolomics should be utilized to accelerate the process of drug discovery and development from plants used traditionally for the treatment of various ailments.

*Vachellia* species are widely distributed over Africa and South America, but originate from the region of the Caribbean according to the boreotropical hypothesis of the historical biogeography of *Vachellia* [3]. They are characterized by their capitate inflorescence and spinescent stipules, which distinguish them from the genus *Senegalia* [4]. The African species of *Vachellia* include *V. karroo*, *V. nilotica*, *V. xanthophloea* and *V. sieberiana* [5]. Medicinal uses of *Vachellia* species encompass the treatment of diarrhea, fever, sore throat and diabetes, among others (Table 1). Several metabolites produced by *Vachellia* species, including saponins, tannins, phenols, flavonoids (apigenin, luteolin and quercetin), sugars, anthocyanins, terpenoids, lactones, polyphenols and amino acids, have been reported to contribute positively to in vitro biological activities [6]. Primary and secondary metabolites may be classified as pro-oxidants or oxidants which are significant in stimulating reactive oxygen species (ROS), that play a role in the pathogenesis of various diseases [7]. Natural antioxidants and pro-oxidants in a daily diet and in traditional medicines provide health benefits, and for this reason, extensive research is required to discover natural sources that possess the reducing power of reactive oxygen/nitrogen compounds [8]. Free radical scavenging activity of metabolites from various *Vachellia* species has previously been reported [9,10,11], however, there is limited literature on the identification of the compounds involved. One approach to comprehensively determine metabolite profiles and to correlate them to activity is by using metabolomic fingerprinting [12]. Metabolomics allows a quantitative and qualitative analysis of complex mixtures of metabolites in an organism [13]. This technique is preferred since it offers diverse chemical profiles and tools that are relevant in a range of applications, such as toxicology, pharmaceutical drug development [14], genotyping [15] and nutrition and environmental assessment [16]. Coupling of spectrometric analytical methods such as liquid chromatography/mass spectrometry (LC-MS), gas chromatography/mass spectrometry (GC-MS) and nuclear magnetic resonance (NMR) to study metabolomics is a powerful strategy that can be used to obtain a comprehensive analysis of metabolites and assist in compound identification [14]. NMR is mainly preferred as a profiling method to study metabolomics since it is rapid and reproducible [17]. However, a shortcoming of NMR is its low sensitivity compared to mass spectrometry, and hence there is no single best method to study metabolomics and all spectrometric methods are complimentary.

Abdel-Farid et al. [18] profiled the variations of metabolites in different organs of *Vachellia* species (*A. nilotica*, *A. seyal* and *A. laeta*), and some secondary metabolites were characterized and correlated to the DPPH scavenging activity. However, no studies have investigated or documented the phytochemistry and biological activities of the different *Vachellia* species that occur in South Africa. The aim of this study was therefore to determine variations of metabolites in these five *Vachellia* species and to correlate metabolites to the antioxidant activity using a chemometric/metabolomics approach coupled with multivariate statistical modeling tools such as PCA and OPLS-DA.

## 2. Results

### 2.1. DPPH Radical Scavenging Activities

The DPPH radical scavenging assay is a well-accepted method to evaluate antioxidant activity. This method is based on the reduction of a purple-colored DPPH radical to a yellow-colored DPPH. The *V. Karroo, V. kosiensis* and *V. xanthophloea* extracts exhibited high radical scavenging activity with IC_50_ values of 4.94 ± 0.44, 5.13 ± 0.40 and 4.91 ± 0.42 µg/mL, respectively. *Vachellia sieberiana* and *V. tortilis* displayed significantly lower DPPH scavenging activity with an IC_50_ of 87.41 and 70.01 µg/mL, respectively.

### 2.2. ABTS^+^ Radical Scavenging Activity

*Vachellia karroo, V. kosiensis* and *V. xanthophloea* exhibited the ability to reduce the ABTS^+^ radical to 50%, with IC_50_ values of 2.23 ± 0.51, 5.61 ± 0.63 and 3.61 ± 0.45 µg/mL, respectively. These results are presented as IC_50_ values for all tested samples (Table 2). Using both methods, *V. karroo* remained the extract that exhibited the strongest radical scavenging capacity.

### 2.3. TLC Bioautographic Radical Scavenging Activities

Evaluation of the radical scavenging activity of *Vachellia* species by thin-layer chromatography (TLC) bioautography demonstrated clear spots interpreted as inhibition zones on a purple or blue-green background, representing non-active extracts. Out of the three solvent systems investigated to provide superior separation of compounds on a TLC plate, the ethyl-acetate:formic acid:water (ratio: 8:1:1) was the best mobile phase used as it exhibited at least 70% separation of compounds. Results obtained using the TLC bioautography method were consistent with findings observed in the 96-well serial dilution method. As observed using the TLC bioautography method, *V. karroo*, *V. kosiensis* and *V.*
*xanthophloea* showed high DPPH and ABTS^+^ scavenging activity as compared to *V. sieberiana* and *V. tortilis* (Figure 1), viewed under UV-light under a short wavelength (254 nm, Figure 1a) and a long wavelength (365 nm, Figure 1b). Several fluorescent compounds were observed under a long wavelength in the *V. sieberiana* and *V. tortilis* extracts, although these compounds accounted for less activity. The advantage of using the TLC bioautography method is that it allows analysis of color intensity, which may vary between samples, and may be interpreted as higher to lower activity.

### 2.4. Ferric Reduction Antioxidant Power (FRAP) Assay

The reducing power of the extracts changed the yellow ferric tripyridyl triazine (Fe^3+^ TPTZ) to deep blue ferrous (Fe^2+^) depending on the reduction potential of the extracts. Results revealed a trend of inhibitory activity of FRAP which increased as the extract concentrations increased. Similar to the DPPH and ABTS^+^ results, all extracts displayed inhibitory activity of FRAP, with *V. kosiensis*, *V. xanthophloea* and *V. karroo* exhibiting greater reduction antioxidant power with IC_50_ results recorded as 11.50 ± 0.34, 23.20 ± 0.34 and 28.14 ± 0.44 µg/mL, respectively. The antioxidant activities of five *Vachellia* extracts showed a concentration-dependent response when tested against the DPPH, ABTS^+^ and FRAP, with a decrease in free radicals at gradual increasing concentrations (Appendix A).

### 2.5. ^1^H NMR Metabolomic Profiles and Antioxidant Activity Correlation

^1^H NMR data were analyzed using SIMCA-P software, and PCA and OPLS-DA plots were constructed to compare the chemical profiles of the plants. PCA is an unsupervised pattern recognition method and was used to ensure an unbiased elucidation of the analyses of the samples. The PC1 and PC2 components together accounted for 95% of the total variance. As shown in Figure 2a, there were three separate groups between the five (50% methanolic extracts) *Vachellia* species. The first cluster represented *V. karroo* and *V. xanthophloea,* cluster 2 represented *V. sieberiana* and *V. tortilis,* while *V*. *kosiensis* samples grouped together to form cluster 3. The OPLS-DA, which is a supervised pattern recognition method, was used to discriminate between samples based on their DPPH scavenging activity. The OPLS-DA (Figure 2b) further separated samples into two clusters, with *V. karroo, V. kosiensis* and *V. xanthophloea* defined as cluster 4, and cluster 5 defined by *V. tortilis* and *V. sieberiana*. Cluster 5 consisted of plant samples that showed less DPPH scavenging activity, in contrast to cluster 4, which showed significantly higher DPPH scavenging activity. The OPLS-DA model had a goodness of fit (R^2^Y) of 0.993 with predictability (Q^2^Y) of 0.989. A value greater than 0.5 indicates that the model possessed good predictive power [31]. A permutation plot was created to validate the model, and the permutation intercepts were R2 = (0.0, 0.269), Q2 = (0.0, −0.674) (shown in Appendix A). Goodness of fit of the OPLS-DA model was further confirmed by cross-validated ANOVA (CV ANOVA), and a significant *p*-value (1.23 × 10^−16^) of the CV ANOVA score, which suggests the robustness of the model (Appendix A). A receiver operating characteristic curve (ROC) and area under the curve (AUC) performance measurement tool was also developed to assess the probability accuracy of the model, which yielded values of 0.994 and 0.945. A ROC (AUC) value greater than 0.8 is considered excellent [32]. The ^1^H NMR spectra of the leaf extracts (Figure 3) were stacked to observe peaks representing metabolites that contribute to the variability of these plants. From the spectra, the samples with high DPPH activity (*V. karroo, V. kosiensis* and *V. xantophloea*) exhibited a very similar profile for the aromatic region δ 6.0–7.0 ppm, whereas the region δ 3.5–4.0 ppm is generally dominated by high levels of sugars and aliphatic compounds in the region δ 0.5–3.0 ppm for the samples with low DPPH activity (*V. sieberiana* and *V. tortilis*).

The ^1^H NMR spectra of the extracts can be classified into 3 regions based on the chemical shift, which are the aliphatic region (0–3 ppm), the sugar region (3–6 ppm) and the aromatic region (6–10 ppm). Figure 3 shows the stacked ^1^H NMR spectra of five *Vachellia* extracts with prominent signal peaks of the sugar region in all samples, which are characteristic peaks of glucose, fructose or sucrose. The second most abundant was in the aliphatic region, especially in the less active *V. sieberiana* and *V. tortilis* extracts. This region displays characteristic peaks of carboxylic acid-type compounds such as alanine, leucine, valine, proline and succinic acid. Relatively less peaks were observed in the aromatic region; however, a few clear similar peaks are present in the region of 6.4–7.6, especially in the samples with high activity. Other interesting chemicals shifts of compounds such as trigonelline (*δ* 9.15 ppm) and formic acid (*δ* 8.42 ppm) were also observed in the less active samples.

^1^H NMR regions which influence the separation of samples were observed on the S-plot and variable importance of projection (VIP) scores of the OPLS-DA (Figure 4a,b). Scores on the two ends of the S-plot and VIP ≥ 1.0 with a significance of *p* < 0.05 ANOVA were selected as influential scores responsible for the separation of samples [33]. The ^1^H NMR regions contributing to the separation of samples were shown on the contribution plot (Figure 5) as 2.48, 2.52, 2.72, 2.76, 3.32, 3.6, 4.32, 4.8 and 4.82 ppm, and to a lesser extent 5.9, 6.0, 6.2, 6.4, 6.6, 7.0 and 7.5 ppm, and the negatively associated regions were shown to be 0.92, 0.96, 1, 1.04, 1.08, 1.28, 1.4, 1.44, 1.84, 1.88, 1.92, 2, 2.04, 2.08, 2.12, 2.16, 2.2, 2.36, 2.40, 2.84, 2.88, 2.96, 3, 3.24, 3.28, 3.36, 3.4, 3.44, 3.48, 3.52, 3.64, 3.68, 3.72, 3.76, 3.84, 3.88, 3.92, 4, 4.08, 4.12, 4.48, 4.6, 4.64 and 5.44 ppm. These regions correlated with buckets on the extreme ends of the S-plot and VIP score values ≥ 1.0.

The ^1^H NMR buckets obtained from the contribution loading plot (Figure 5) representing compounds of importance that contribute to the clustering of samples were annotated through online network database (HMDB and Chenomx library) searches. Possible annotation and chemical structures of the metabolites were preliminarily evaluated. Metabolite variation between the two sample clusters 4 (active) and 5 (inactive) of the OPLS-DA model (Figure 2) associated with antioxidant activity were assigned and are reported in Table 3.

### 2.6. UHPLC-qTOF-MS Analysis Results

The tentative identification of the metabolite composition of five *Vachellia* 80% methanolic extracts was quantitatively performed by ultra-high-performance liquid chromatography–quadrupole time-of-flight mass spectrometry (UHPLC-qTOF-MS) in a negative and positive ionization mode. From the results, as shown in Table 4 and Figure 6, it is evident that a class of polyphenolic compounds characterized by the di- or tri-hydroxyl groups on the B-ring and the meta-5,7-dihydroxyl groups on the A-ring were the prevailing phytoconstituents of this analysis. Catechin (**1**), epicatechin (**2**) and epigallocatechin (**3**) were tentatively identified at *m/z* 290.07904, 290.07904 and 306.07395, respectively. The latter compounds together with epigallocatechin gallate (EGCG) (**5**) shown by a peak at *m/z* 305.0659 are typical phenolic compounds with potent antioxidant activity abundant in tea, cocoa, berries and wine [34]. Additionally, gallic acid and methyl gallate were annotated at *m/z* 169.0133 and 183.0294, respectively. Phenolic compounds containing a flavonoid moiety were also annotated, and these include luteolin rutinoside (*m/z* = 609.1467), kaempferol (*m/z* = 285.0398) as well as chrysoeriol rutinoside (*m/z* = 607.1663). Baicalein (**17**) and its glucuronide conjugate baicalin (**16**) flavonoids were shown by the presence of peaks at *m/z* 269.0443 and 445.0748, respectively. Most important was the annotation of epigallocatechin, methyl gallate and quercetin, which are present in all highly active extracts (*V. karroo, V. kosiensis* and *V. xanthophloea*) and may be responsible for the enhanced antioxidant activity observed. Other compounds detected include luteolin glucoside (**13**), luteolin rutinoside (**10**) and cyanidin rhamnoside (**12**), which are renowned as strong inhibitors of DPPH, ABTS^+^ and FRAP radicals [27,28].

## 3. Discussion

The DPPH, ABTS^+^ and FRAP results obtained in this study revealed that the *Vachellia* species tested contain potent antioxidants, although in this case, the scavenging activities of the extracts were lower than ascorbic acid, with the lowest IC_50_ values being indicative of higher antioxidant capacity in the tested extracts. By comparing the antioxidant methods used in this study, it can be inferred that the quantified antioxidant activity was not equivalent, and the ABTS^+^ method displayed a lower sensitivity compared to the DPPH method. Tshikalange et al. [35] reported that the antioxidant activity of the ethanolic root extracts of *V. karroo* exhibited DPPH scavenging activity with an EC_50_ of 0.83 µg/mL, with the positive control (ascorbic acid) exhibiting an EC_50_ of 1.44 µg/mL. The ability of the *V. karroo* aqueous and acetone extracts to scavenge DPPH and FRAP radicals has been previously reported as IC_50_ = 0.67, 0.62 mg/mL for DPPH and 0.59, 0.56 mg/mL for FRAP [36]. A comparative study of the antioxidant activity of *Vachellia* species grown in Saudi Arabia using the DPPH scavenging method showed that *V. tortilis* demonstrated an EC_50_ of 250.13 μg/mL [37]. An investigation of *V. nilotica* showed high radical scavenging activity (IC_50_ = 6.5 μg/mL) when the leaves were sequentially extracted with ethanol, although extraction with non-polar solvents (petroleum ether, benzene and dichloromethane) displayed lower DPPH scavenging activity [38]. A previous study by Katerere et al. [7] reported the TLC bioautographic antioxidant activity of acetone extracts of *V. galpinii*, *V. karroo*, *V. xanthophloea* and *V. sieberiana*, where the authors observed three compounds that exhibited strong free radical scavenging capacity. The latter study also demonstrated that chloroform extracts displayed less DPPH scavenging activity as compared to acetone extracts. Similar challenges to those experienced in our study of finding a suitable solvent system for separation of compounds on a TLC plate were also encountered by Katerere et al. [7]. A poor separation of compounds was observed with a moderately polar acidic solvent system: chloroform:ethylacetate:formic acid (CEF: 20:16:4) in all the extracts, and a good separation was obtained with a more polar solvent system: ethylacetate:methanol:water (EMW: 40:5.4:4). DPPH free radical scavenging of *V. xanthophloea*, *V. tortilis, V. nilotica* and *V. nigrescens* displayed an IC_50_ value ranging from 0.57 to 0.76 µg/mL, compared to the positive control (ascorbic acid) with an IC_50_ of 0.48 µg/mL [34]. The high antioxidant activity observed in our study and previously published studies may be due to the presence of phenolic compounds found in *Vachellia* species [39,40]. Additionally, compounds in *V. karoo*, *V. kosiensis* and *V. xanthophloea* extracts displaying high DPPH and ABTS^+^ scavenging activity suggests that the mechanisms of action of these compounds may be the same for both methods, including either hydrogen atom transfer or single electron transfer, or a combination of both mechanisms.

The main objective of this study was to provide a complete phytochemical profile of five *Vachellia* species that are responsible for reducing free radicals. This was mainly achieved by exposing extracts to ^1^H NMR spectroscopy and UHPLC-qTOF-MS analyses, which have been extensively used for profiling of metabolites from plants [18,39]. Due to its high reproducibility, non-destructive and non-invasive nature, NMR was suitable for metabolite fingerprinting. However, even though ^1^H NMR permits a rapid and accurate quantification of metabolite analysis, detection of metabolites presented in low levels required the use of a more sensitive analytical technique, such as UHPLC-qTOF-MS [17]. The UHPLC-qTOF-MS analysis revealed compounds that were responsible for the separation of *Vachellia* spp. into potent and less active groups. However, the presence of epigallocatechin, methyl gallate and quercetin, which were only present in the three most active samples (*V. karroo*, *V. kosiensis* and *V. xanthophloea*), was postulated as being responsible for the high activity observed. The presence of these compounds is also supported by the ^1^H NMR analysis with the characteristic peaks of the catechin compounds observed around 6.0 and 7.0 ppm, quercetin at 7.0 and 7.5 ppm and methyl gallate at 3.7 and 7.1 ppm in the ^1^H NMR spectra. Phytochemical screening of different parts of *V. nilotica* have previously revealed the presence of saponins, tannins, alkaloids, flavonoids and phenols in the ethanol extracts of the leaves, and further investigation utilizing LC-ESI-MS/MS demonstrated the presence of epicatechin-5-gallate, epicatechin, methyl gallate, gallic acid, vitexin, L-arabinose and caffeic acid hexose from the leaf samples of *V. nilotica* [35]. Isolation of phytochemicals from *Vachellia* species led to the identification of flavonoids (apigenin, luteolin and quercetin) from the leaves of *V. tortilis*, and the presence of apigenin glycoside, quercetin glycoside and isorhamnetin glycoside from leaves, while *n*-hexacosanol, betulin, α-, β-amyrin and β-sitosterol from the stem bark has also been previously reported [36]. Our UHPLC-qTOF-MS analysis is in agreement with an earlier study where epicatechin, β-sitosterol and epigallocatechin were isolated from leaf extracts of *V. karroo*, which demonstrated activity against *Listeria monocytogenes* [11]. According to the literature, it is evident that the aforementioned metabolites possess antioxidant activities. In vivo studies have demonstrated that epigallocatechin, methyl gallate and quercetin display radical scavenging effects, reduce oxidative stress and exhibit cytoprotective effects on human keratinocytes (HatCaT) and murine (RAW264.7) cell lines [41]. The isolation of compounds from methanol leaf extracts of *V. sieberiana* led to the identification of eight polyphenols, including ellagic acid, gallic acid, isoferulic acid, quercetin, kaempferol, quercetin 3-O-β-D-glucoside, kaempferol-3-alpha-L-arabinoside and 6,7,8-trihydroxy-3,4’-dimethoxydihydroflavone [42]. Xu et al. [43] reported that the antioxidant activities of catechins are closely related to their molecular structures, especially the hydroxyl and galloyl groups.

Despite the lack of published literature on the pharmacological activities of some *Vachellia* species, especially antioxidant activity, our results demonstrate that *V. kosiensis* and *V. sieberiana* both exhibit antioxidant activity, with our study also elucidating the metabolites that separated the *Vachellia* species from each other and demonstrating their possible free radical scavenging activities. These results therefore warrant further investigation to assess the antioxidative phytochemicals.

## 4. Materials and Methods

### 4.1. Plant Collection

Five *Vachellia* species, namely *V. sieberiana, V. tortilis, V. karroo, V. kosiensis* and *V. xanthophloea,* were collected into paper bags from the campus of the University of Pretoria and immediately taken to the laboratory to be dried. Five samples of each species were collected from five trees. Plant materials were identified, and voucher specimens were deposited in the H. G. W. J. Schweickerdt Herbarium (PRU), University of Pretoria.

### 4.2. Sample Preparation

#### 4.2.1. Plant Extraction

Leaves were dried at room temperature and pulverized to a fine powder using a grinding mill (IKA^®^ MF10, Munich, Germany). Fifty grams (50 g) of powdered plant material were extracted with 500 mL of 80% methanol-water. Samples were filtered using the Buchi^®^ filtration system (Sigma-Aldrich^®^, Darmstadt, Germany), and a Stuart^®^ rotary evaporator (IKA^®^, Munich, Germany). was used to concentrate the extracts. Extracted samples were further lyophilized to dryness.

#### 4.2.2. Sample Preparation for ^1^H NMR Measurement

Pulverized plant samples (50 mg) were weighed into a 2 mL Eppendorf tube and extracted using 1500 µL deuterated methanol-water (1:1) with potassium phosphate monobasic (KH_2_PO_4_) buffer in deuterium oxide (D_2_O) and 0.1% trimethylsilypropionic acid sodium salt (TSP). The buffer was adjusted to pH 6.0 using 1N deuterated sodium hydroxide (NaOD). This extraction was performed to target the polar constituents of the plants. Samples were sonicated for 60 min (min), centrifuged for 15 min and the supernatant was filtered through a 0.22 µm syringe filter. All samples were transferred to 5 mm NMR tubes for ^1^H NMR analyses on a Varian NMR 600 MHz spectrometer (Varian Inc., Palo Alto, CA, USA) with consistent settings, and 32 scans were recorded. Samples were replicated five times.

#### 4.2.3. Sample Preparation UHPLC-qTOF-MS Measurement

Extraction of metabolites from the leaves was performed following the method of [33] with slight modifications. Dried powdered leaf samples (50 mg) were suspended in 80% methanol/water, sonicated for 15 min and centrifuged for 15 min at 3000× *g* at 4 °C. The supernatants were filtered through a 0.22 µm nylon syringe filter and evaporated to 1 mL using an EZ-2 plus evaporator (GeneVac™, St. Louis, MO, USA). Samples were stored in glass vials for NMR and LC/MS analysis.

#### 4.2.4. 2,2-diphenyl-2-picryl-hydraxyl (DPPH) Free Radical Scavenging Activity Assay

The DPPH free radical scavenging activity method by Rangkadilok et al. [44] with slight modification was followed. Briefly, 20 μL of 10,000 μg/mL of extract was pipetted into 180 μL of methanol and added to the top row of a 96-well plate. The solutions were serially diluted onto the remaining wells of the 96-well plate, which contained 100 μL of methanol. One hundred microliters of 90 μM methanolic DPPH was added to all the wells. The final concentrations of the extract ranged from 3.9 to 500 μg/mL. The plates were incubated for 30 min at 25 °C and the absorbance was measured on a microplate reader (VarioSkan Flash^®^, Thermo Fisher Scientific, Vantaa, Finland) at a wavelength of 517 nm. Ascorbic acid (vitamin C) was used as the positive control at the same concentrations as the extracts. The DPPH inhibitory percent was calculated using formula (1).
DPPH radical scavenging effect (%) = [(A_0_ − A_t_)/A_0_] × 100%(1)
where A_0_ and A_t_ are the absorbance values of the blank control and tested sample or positive control, respectively. The EC_50_ value represents the 50% inhibition ratio of DPPH activity.

#### 4.2.5. 2,2’-azino-bis(3-ethylbenzothiazoline-6-sulfonic Acid) (ABTS^+^) Radical Cation Scavenging Activity Assay

The ABTS^+^ radical cation scavenging activity assay was assessed using a modified method described by Re et al. [45]. In brief, the ABTS^+^ cation radical was formed by combining 7 mM of ABTS^+^ and 2.45 mM of potassium persulfate (K_2_S_2_O_8_) in distilled water and kept in the dark for 16 h. Thereafter, the optical density (OD) of the mixture was adjusted with methanol to an absorbance of 0.70 (±0.02) at 734 nm. The ABTS^+^ scavenging activity assay was conducted according to the DPPH scavenging activity assay. The absorbance was measured on a microplate reader (Varioskan-Flash^®^, Thermo Fisher Scientific, Vantaa, Finland) at a wavelength of 734 nm. Ascorbic acid was used as the positive control at the same concentrations as the extracts. The ABTS^+^ scavenging activity percentage was calculated using formula (1) shown in Section 4.2.4.

#### 4.2.6. Ferric Reducing Antioxidant Power

The ferric reducing antioxidant power (FRAP) of the extracts was investigated using the FRAP reagent prepared by mixing (10:1:1, *v/v/v*) acetate buffer (300 mM, pH 3.6), tripyridyl triazine (TPTZ) (10 mM in 40 mM of HCl) and FeCl_3_⋅6H_2_O (20 mM), which was mixed with varying extract concentrations (3.9–500 µg/mL). Ascorbic acid was used as a reference standard and all experiments were performed in triplicate. After 30 min incubation at room temperature, the absorbance was determined at a wavelength of 593 nm on a microplate reader (Varioskan-Flash^®^ Flash, Thermo Fisher Scientific, Vantaa, Finland).

#### 4.2.7. TLC Bioautographic Radical Scavenging Assay

The thin-layer chromatography (TLC) was developed to carry out separation and assess the DPPH and ABTS^+^ scavenging activity of five *Vachellia* species in support of the 96-well dilution method. Briefly, extracts were dissolved in methanol and aliquots of 10 µL were spotted on three TLC plates (Merck, silica gel 60 F_254_, 10 × 10 cm). The TLC plates were developed in a pre-saturated solvent chamber with various solvent systems of different polarities to determine the best separation of the compounds. These multiple solvent systems included mixtures (*v/v*) of toluene:ethylacetate:methanol (5:5:1), chloroform:ethylacetate:formic acid (5:5:1) and ethylacetate:formic acid:water (8:1:1) as mobile phase. The mobile phase was allowed to move up on the TLC plate until it reached the 1 cm solvent front line, then the plates were removed from the chamber to air-dry for 30 min. One plate was visualized under a UV-light (wavelengths of 254 and 365 nm) and sprayed with vanillin, after which a heat gun (PROHEAT^®^ Sigma-Aldrich, Master Appliance Corp, Racine, WI, USA) was used to develop the optimal color of the compounds. Other TLC plates were immersed in DPPH and ABTS^+^ solution to observe the radical scavenging activity. The DPPH scavenging activity was indicated by a white-yellow band on a purple background.

#### 4.2.8. ^1^H NMR Measurement and Data Analysis

Spectra of the samples were generated using a 600 MHz NMR operating at a frequency of 599.74. The data were processed using MestReNova software (version 9.0.1, Mestrelab Research, Spain), and phase correction, baseline correction, referencing and normalization with respect to the stable standard trimethylsilyl propionate (TSP) were performed on the ^1^H NMR spectra. The spectral regions from δ = 0.04 to δ = 10.00 were bucketed into 0.04 ppm bins. The residual water (4.6–5.0 ppm) and methanol (3.28–3.32 ppm) solvent peaks were excluded from the analysis. The processed spectral data obtained were exported to Microsoft Excel and transformed to csv files prior to SIMCA processing. The PCA and OPLS-DA multivariate data analyses were conducted with SIMCA-P software (version 13.0, Umetrics, Umea, Sweden), where pareto scaling was used to further transform data. This scaling method is a more stable technique that amplifies variances between samples. The contribution plot, S-plot and VIP plot were contracted to determine the chemical variation between the samples.

#### 4.2.9. UHPLC-qTOF-MS Analysis

The tentative identification of the metabolite composition of five *Vachellia* 80% methanolic extracts was quantitatively performed by ultra-high-performance liquid chromatography–quadrupole time-of-flight mass spectrometry (UHPLC-qTOF-MS) in a negative and positive ionization mode. Pulverized leaves (5 mg) were extracted with 1.5 mL of 80% methanol (LC-grade and ultrapure LC-grade water), homogenized, ultrasonicated for 5 min, and the homogenates were centrifuged for 15 min. The extract of each sample was then filtered using 0.22 µm nylon syringe filters and the filtrates were concentrated by evaporation to dryness. The dried extracts were resuspended with 300 µL of 50% methanol and pipetted into 2 mL HPLC glass vials. Aliquots of extracts were prepared in triplicates and stored at −20 °C before analysis. The chromatographic separation and mass spectrometry detection were performed on a Waters Classic UHPLC coupled in tandem to a Waters SYNAPT G1 HDMS mass spectrometer (Waters, Manchester, UK). An HSS T3 C18 column (150 × 2.1 mm, 1.8 µM), thermostatted at 60 °C, was used to achieve the separation of metabolites. Elution solvents, Eluent A (10 mM formic acid and acetonitrile) and Eluent B (10 mM formic acid), were used at a flow rate of 0.4 mL/min. The initial mobile phase consisted of a gradient combination of Eluents A and B, comprising 98% of Eluent A, maintained for 1 min. The gradient was altered to 5% Eluent A by 25 min. These conditions were maintained for 2 min and thereafter returned to the initial mobile phase conditions. To avoid variations in data, samples were run in triplicate and solvent blanks were included in the run.

The Waters SYNAPT G1 Q-TOF system (Waters, Manchester, UK) was used in V-optics mode to obtain high-resolution mass spectra. Electrospray analysis was performed in positive and negative ionization mode to enable detection of phenolic compounds and other ESI-compatible compounds. Conditions were set as follows: Typical mass accuracies between 1 and 5 mDa were obtained by lock mass calibrant using leucine enkephalin (50 pg/mL) as a reference. The spectrometer was operated in both ESI positive and negative modes with a capillary voltage of 2.5 kV, with the sampling cone at 30 V and the extraction cone at 4.0 V. The source temperature was 120 °C and the desolvation temperature was set at 450 °C. Nitrogen gas was used as the nebulization gas at a flow rate of 550 L h^−1^ and cone gas was added at 50 L h^−1^. MassLynx v4.1 (SCN 872) software (Waters Corporation, Milford, MA, USA) was used to control the hyphenated system and to perform all data manipulation. MassFragment v.2.0.w.15 (Waters Corporation, Milford, MA, USA) was used to evaluate all mass spectra in relation to proposed structures. Metabolite profiling was carried out by comparing the mass spectra, retention time (Rt) and ion fragments with data from various databases, including the NIST (National Institute of Standards and Technology) database, DNP (Dictionary of Natural Products: www.dnp.chemnetbase.com, accessed on 22 June 2021), METLIN Mass Spectral Database, MassBank (USA) and mzCloud (Advanced Mass Spectral Database). In addition, data from peer-reviewed literature were used to check for the annotation of the measured masses.

#### 4.2.10. Statistical Analysis

Experiments were conducted in triplicate and all results are presented as mean ± standard deviation (SD). Experimental data were analyzed using GraphPad Prism software 8.2 (GraphPad Software, CA, USA). For all the analyses, differences were considered statistically significant at *p* < 0.05, determined using the Duncan’s multiple range test. Significantly discriminant metabolites were characterized by VIP > 1 and *p* ≤ 0.05. A ROC curve was generated using only the metabolites with a significant statistical variation, and the area under the curve (AUC) of the ROC analysis of 0.879.

## 5. Conclusions

Among the five tested extracts, *V. karoo*, *V. kosiensis* and *V. xanthophloea* extracts clustered together in the OPLS-DA analysis, indicative of a similar chemical profile and additionally showed the highest free radical scavenging effects. The *V. tortilis* and *V. sieberiana* samples grouped together and displayed less activity against DPPH. The ^1^H NMR- and UHPLC-qTOF-MS-based metabolomics coupled with multivariate statistical tools were used to annotate and identify metabolites contributing to the diversity of the five *Vachellia* species. By using the various analytical techniques, it is possible to reveal compounds responsible for the DPPH scavenging activity as these compounds are common in all the samples. From these plant extracts, metabolites such as epigallocatechin, methyl gallate and quercetin were putatively identified as major constituents of the three most active plants, all known for their strong free radical scavenging effects. The list of compounds also includes various compounds with antioxidant and medicinal activities and provides an insight into the rich biochemical composition of the *Vachellia* species and possible interactive effects of the various chemical components in each plant. The findings of the current work provide strong scientific support toward the applications of analytical techniques such as ^1^H NMR and UHPLC-qTOF-MS to accelerate drug discovery and development. Moreover, ethnobotany coupled with chemometric analytical techniques in the search for therapeutic agents may represent an effective strategy for the identification of leading therapeutic compounds.

## Figures and Tables

**Figure 1 molecules-26-06214-f001:**
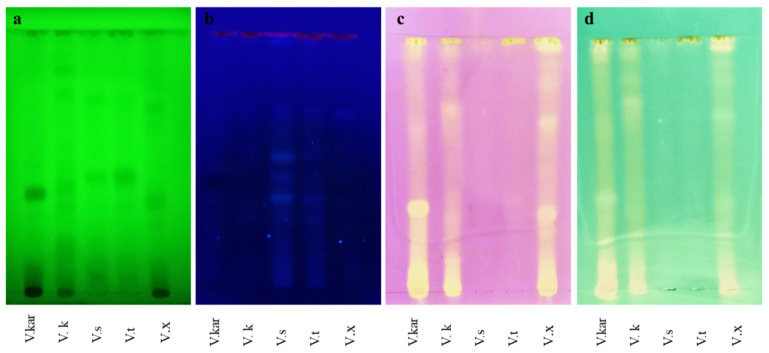
TLC chromatogram of five *Vachellia* extracts showing radical inhibitory compounds visualized using (**a**) UV-light, short wavelength (254 nm), (**b**) UV-light, long wavelength (365 nm), (**c**) DPPH and (**d**) ABTS^+^. Chromatogram was developed on a mobile phase: acetic acid:formic acid:water (8:1:1, *v/v*). Clear spots on a purple/blue-green background indicate inhibitory activity (Vx, *V. xanthophloea*; Vkar, *V. karroo*; Vk, *V. kosiensis*) and less active (Vs, *V. sieberiana*; Vt, *V. tortilis*).

**Figure 2 molecules-26-06214-f002:**
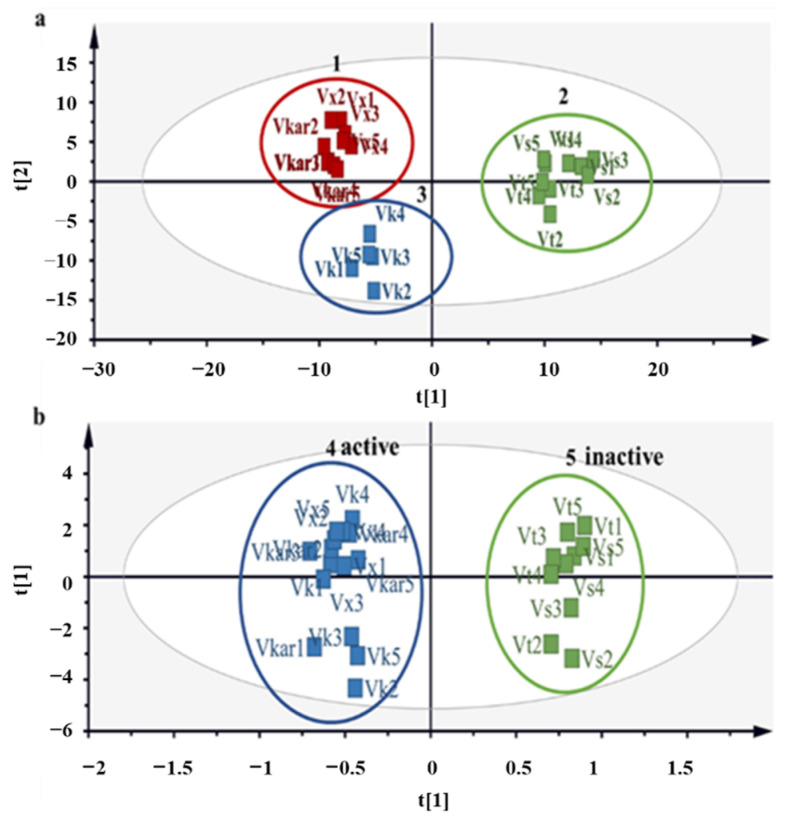
A representation of the PCA score plot (**a**) and OPLS-DA (**b**) score plot showing the separation of the plant extracts with high activity in blue (Vx, *V. xanthophloea*; Vkar, *V. karroo*; Vk, *V. kosiensis*) and less active samples in green (Vs, *V. sieberiana*; Vt, *V. tortilis*). Extracts with IC_50_ ≤ 10 µg/mL are considered active and IC_50_ ≥ 10 µg/mL are deemed inactive.

**Figure 3 molecules-26-06214-f003:**
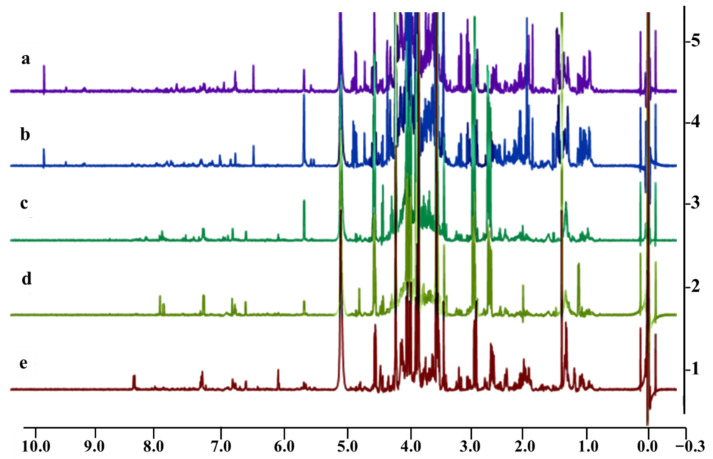
Stacked ^1^H NMR spectra of leaves’ extracts of (**a**) *V. sieberiana,* (**b**) *V. tortilis,* (**c**) *V. kosiensis,* (**d**) *V. xanthophloea* and (**e**) *V. karroo*.

**Figure 4 molecules-26-06214-f004:**
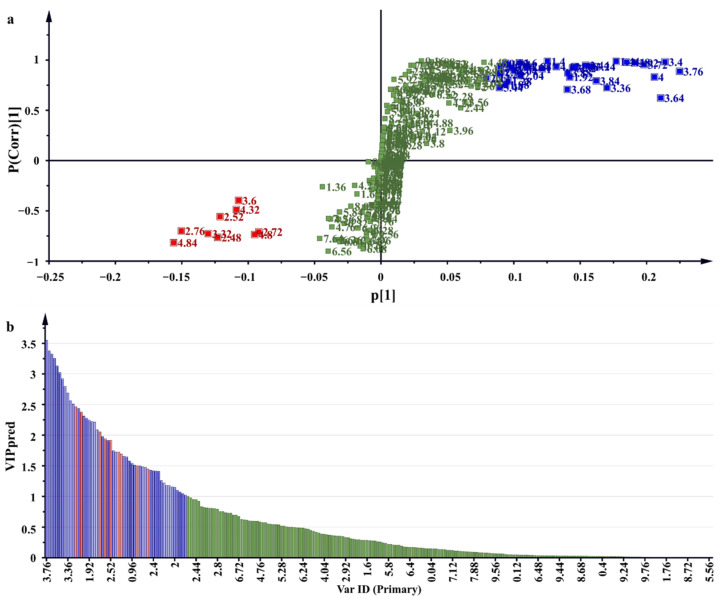
A representative OPLS-DA-derived S-plot (**a**) and VIP score plot (**b**) showing ^1^H NMR regions that contributed to clustering and separation of samples. Cluster (**a**) ^1^H NMR regions with high DPPH scavenging activity in red and cluster (**b**) ^1^H NMR regions associated with low DPPH scavenging activity in blue.

**Figure 5 molecules-26-06214-f005:**
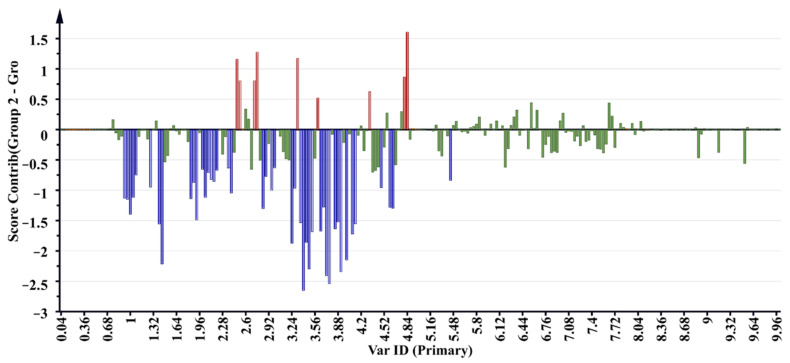
Contribution loading plot of samples representing regions responsible for the clustering of the samples, with bars above the line positively associated and bars below the line negatively associated with the activity.

**Figure 6 molecules-26-06214-f006:**
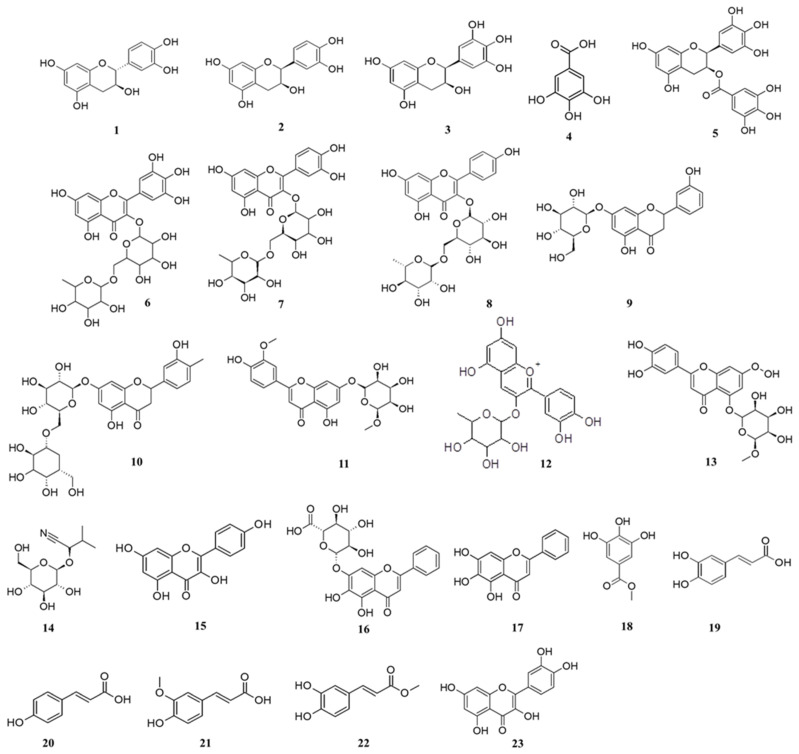
Chemical structures of metabolites annotated from *Vachellia* species.

**Table 1 molecules-26-06214-t001:** Medicinal uses of *Vachellia* species in African countries.

Plant Name	Medicinal Uses	Plant Parts Used	Country of Use/ References
*V. sieberiana*	Convulsions, sedative (mental illness), dizziness	Bark	Zimbabwe [19]
	Fever	Leaves, Bark, Roots	South Africa [20]
	Sore throat	Root	Nigeria [21]
	Diarrhea	Bark	South Africa [22]
*V. xanthophloea*	Diabetes	Bark	Zimbabwe [19]
	Emetic/cathartic, sickle cell anemia	Roots	Tanzania [23]
	Fatigue, indigestion, skin disorders	Bark, Roots	Kenya [24]
	Febrile, fevers, gingivitis, high cholesterol	Leaves, Bark, Roots	South Africa [25]
	Malaria, emetic, mouth sores, pharyngitis, tuberculosis symptoms	Bark	South Africa [20]
*V. karroo*	Diarrhea, dysentery, gastrointestinal, venereal diseases	Leaves, Roots	South Africa [25,26]
	Fractures, diarrhea	Bark	South Africa [27]
	aphrodisiac, sexually transmitted infections, urinary schistosomiasis	Bark, Roots	Zimbabwe [27,28]
*V. kosiensis*	No literature on the medicinal uses found		
*V. tortilis*	Cough	Bark	Nigeria [21]
	Stomach-ache, digestive	Fruits	Yemen [29]
	indigestion, malaria, strengthen bones, kidney cleanser	Roots	Kenya [30]
	Diarrhea	Branch tips	South Africa [27]

**Table 2 molecules-26-06214-t002:** Inhibitory concentration (IC_50_) values of antioxidant activity of five *Vachellia* species and ascorbic acid (positive control). The IC_50_ (µg/mL) values are expressed as the mean ± standard error (*n* = 3). Bold values are considered noteworthy.

Plant Extracts	DPPH (IC_50_ µg/mL)	ABTS^+^ (IC_50_ µg/mL)	FRAP (IC_50_ µg/mL)
*V. karroo*	**4.94 ± 0.44**	**2.23 ± 0.51**	28.14 ± 0.44
*V. kosiensis*	**5.13 ± 0.40**	**5.61 ± 0.63**	**11.50 ± 0.34**
*V. sieberiana*	87.41 ± 0.58	52.03 ± 0.59	107.09 ± 0.64
*V. tortilis*	70.01 ± 0.30	45 ± 0.58	97.44 ± 0.54
*V. xanthophloea*	**4.91 ± 0.42**	**3.61 ± 0.45**	23.20 ± 0.34
Ascorbic acid	**1.40 ± 0.21**	**1.10 ± 0.20**	**15.10 ± 0.39**

DPPH = 2,2-diphenyl-1-picrylhydrazyl; ABTS^+^ = 2,2’-Azino-bis (3-ethylbenzthiazoline-6-sulfonic acid); IC_50_ value = inhibitory concentration of antioxidants that decrease the radical concentration by 50%.

**Table 3 molecules-26-06214-t003:** ^1^H NMR assignment (characteristic ^1^H NMR peaks) of some compounds contributing to the differentiation of the active and inactive *Vachellia* extracts obtained using Chenomx software and the HMDB database. S = singlet; d = doublet; m, multiplet.

Compounds	δ1H (ppm) and Multiplicity	Active/Inactive
Isoleucine	0.92 (d)	Inactive
leucine	0.96 (d)	Inactive
Valine	1.04 (d)	Inactive
Alanine	1.44 (d)	Inactive
Acetate	1.92 (s)	Inactive
Succinate	2.40 (s)	Active
Citric acid	2.52 (d)	Active
Aspartate	2.70 (m)	Inactive
Choline	3.24 (s)	Inactive
Betaine	3.32 (s)	Inactive
Glucose	4.84 (m)	Inactive
Sucrose	5.44 (d)	Inactive
Catechin	5.92 (s)	Active
Ferulate	6.44 (s)	Active
Gallate	7.10 (s)	Active
Trigonelline	9.15 (s)	Inactive

**Table 4 molecules-26-06214-t004:** UHPLC-qTOF-MS metabolites from the methanol-water leaf extracts of five different *Vachellia* species.

No.	Tentative Metabolites	Empirical Formula	Detected Mass (*m/z*)	Mass Accuracy (mDa)	MS/MS Fragmentation ions	Data Source	*Vachellia karroo*	*Vachellia kosiensis*	*Vachellia tortilis*	*Vachellia sieberiana*	*Vachellia xanthophloea*
1	Catechin	C_15_H_14_O_6_	289.0705	−0.7	Trace	NIST 2014	✓	✕	✕	✕	✕
2	Epicatechin	C_15_H_14_O_6_	289.0716	0.4	Trace	NIST 2014	✓	✕	✕	✕	✕
3	Epigallocatechin	C_15_H_14_O_7_	305.0659	−0.2	Trace	NIST 2014	✓	✓	✕	✕	✓
4	Gallic acid	C_7_H_6_O_5_	169.0133	−0.4	125.02	NIST 2014	✕	✓	✕	✕	✓
5	Epigallocatechin gallate	C_22_H_18_O_11_	457.0786	1.5	169.02	NIST 2014	✕	✕	✕	✕	#
6	Myricetin rutinoside	C_27_H_30_O_17_	625.1419	1.4	316.02	NDP	✓	✓	✓	✕	✓
7	Rutin	C_27_H_30_O_16_	609.1454	−0.2	300.03	NIST 2014	✓ *	✓	✕	✕	✕
8	Kaempferol rutinoside (Nicotiflorin)	C_27_H_30_O_15_	593.1498	−0.8	285.04	NIST 2014	✓	✓	✓	✓	✕
9	Chrysoeriol rutinoside	C_28_H_32_O_15_	607.1663	−1.6	461.10; 299.05	NDP	✕	✕	✓	✓	✕
10	Quercetin rutinoside	C_27_H_30_O_16_	609.1467	1.1	300.03	NIST 2014	✓ *	✓	✓	✓	✓
11	Chrysoeriol glucopyranoside	C_22_H_22_O_11_	461.1072	−1.2	299.02	NDP	✕	✓	✓	✓	✓
12	Cyanidin rhamnoside	C_21_H_20_O_10_	431.0989	1.1	285.03	Metlin DB	✕	✓	✓	✓	✓
13	Luteolin glucoside	C_21_H_20_O_11_	449.1096	1.2	287.05	NIST 2014	✕	✓	✓	✓	✓
14	Dihydroacacipetalin	C_11_H_19_NO_6_	262.1281	0.2	Trace	NDP	✕	#	✓	✓	✓
15	Kaempferol	C_15_H_10_O_6_	285.0398	−0.1	151.00; 107.01	NIST 2014	§	✓	✓	✓	✓
16	Baicalin	C_21_H_18_O_11_	445.0748	−2.3	Trace	NDP	✕	*	✕	#	✕
17	Baicalein	C_15_H_10_O_5_	269.0443	−0.7	Trace	NDP	✓	#	✓	✓	#
18	Methyl gallate	C_8_H_8_O_5_	183.0294	0.0	183.02; 124.03	NIST 2014	✓	✓	✕	✕	✓
19	Caffeic acid	C_9_H_8_O_4_	179.0319	−2.5	Trace	NDP	✕	✕	✕	✕	#
20	*p*-coumaric acid	C_9_H_8_O_3_	163.0394	−0.1	119.04	NIST 2014	#	✕	§	§	✕
21	Ferulic acid	C_10_H_10_O_4_	193.0491	−1.0	Trace	NDP	§ *	§ *	§ *	§ *	✕
22	Methyl caffeate	C_10_H_10_O_4_	193.0501	−1.9	Trace	NDP	§ *	§ *	§ *	§ *	✕
23	Quercetin	C_15_H_10_O_7_	301.0358	1.0	179.00; 151.00	NIST 2014	✓	✓	✕	✕	✓

§ = Observed as a product ion; * = Uncertainty due to similar spectra; # = Trace level concentrations; ✓ = present; ✕ = absent. No MS/MS ions for compounds present at trace levels. Bolded are metabolites present in most of the active extracts.

## Data Availability

The raw data supporting the findings presented in this study are available in the Mendeley repository: https://data.mendeley.com/library, accessed on 24 September 2021 (Garland kgosi, 2021), doi:10.17632/ghn8ffzsnc.1.

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
