# Peer review of "Metabolomic Profiling of Antioxidant Compounds in Five Vachellia Species"

_molecules, 2021, doi:10.3390/molecules26206214_

Round 1
Reviewer 1 Report
Metabolomic Profiling of Antioxidant Compounds in Five 2 Vachellia Species
The authors explored the chemical variations in five species of Vachellia and correlated metabolite profiles (obtained by H NMR) to their antioxidant activity measured by three different methods. Furthermore, metabolites were annotated based on LC-MS/MS data. The manuscript is well written, and the methodology followed is accurate. Something that needs to be praised is the validation of the OPLS-DA model in the software SIMCA, which was comprehensive and well-performed. However, there is several missing details that precluded a better assessment of the manuscript and compromise its quality, thus requiring a major review. Some of these details are:
- NMR processing: Please include all details relative to the data processing. For example, was phase and baseline corrections applied in MestreNova? Did they apply any type of normalization while exporting the data from MestreNova?
- After exporting the data from MestreNova, did they eliminate the peaks relative to the solvent?
- Once in SIMCA, which normalization method was applied and what was the criteria to select it?
- If the OPLS-DA used the DPPH results as Y variable, I suggest labelling cluster 4 and 5 as “active” and “inactive” or something similar to ease interpretation. Maybe defining a threshold of what they can consider “active” and “inactive” can help.
- While the selection of the discriminant metabolites for the “active” and “inactive” samples was made rigorously, I believe the authors should have linked the chemical shifts in the “active” and “inactive” samples to specific metabolites. In table 3, they attempted to do something like this, but it is not clear which shifts are specific of the “active” samples and which in the “inactive”. Maybe adding an additional column to that table with their respective category will help in that.
- In the methods section there were no details relative to the identification of metabolites. I suggest including lines 227 to 231 in the methods. I also suggest adopting the minimum standards in the “Metabolomics Standards Initiative” in the accuracy of the identifications.
- In the LC-MS section the authors should correct m/z values of the reported metabolites with the experimental value obtained (including decimal places as they were obtained from a high-resolution MS). In its current form the authors reported the neutral mass of the metabolites which does not represent the m/z value as m/z (mass-over-charge) implies that the molecule is ionized, which in ESI MS is achieved by gaining of losing a proton (or other adducts), so the mass is different. This mistake is quite surprising from an overall good quality paper. I also suggest including an image with the LC-MS chromatogram of a representative sample from the active and inactive samples to assess visual differences.
- I have serious concerns about the identity of the metabolites reported in Table 4 as no m/z values were reported (current values represent neutral masses) and no MS2 data was provided. This information is key to assess the accuracy in the identifications. The error in ppm between the theoretical and experimental values should also be provided. Furthermore, the glycosylation position of some flavonoids was declared but no information was provided as to how was this glycosylation position determined based on MS data. Please clarify. Furthermore, the authors differentiated some isomeric compounds based on MS data in table 4 but no information was provided as to how was this differentiation made.
- Raw LC-MS and NMR files should be made publicly available in an online repository, so that the science community can assess their data.
- If the authors analysed their data by NMR and LC-MS, why they didn’t attempt to apply the same multivariate methods used for NMR to the LC-MS data? Or even better, they could have concatenated both datasets to have a better representation of the sample’s metabolome and its correlation with antioxidant activity.
Minor points:
Merge items 4.2.9 and 4.2.10 as both refer to LC-MS parameters
Line 23 – tools such as…
Author Response
We would like to thank reviewer 1 for the valuable feedback regarding the comment regarding the manuscript and interpretation of data presented in our study

Reviewer 2 Report
The manuscript entitled “Metabolomic Profiling of Antioxidant Compounds in Five Vachellia Species” analyzed and compared the antioxidant activity of five different Vachiella species from South Africa using different in vitro assays (DPPH, ABTS, FRAP, TLC) and investigated their metabolites profile through 1HNMR and UHPLC-qTOF-M.
The manuscript is quite original and enriches literature data about metabolites and antioxidant profile of Vachiella species from South Africa not yet analyzed.
Major revisions
Abstract
- Revise the abstract including the TLC assay and explain the UHPLC-qTOF-MS acronym.
Introduction
- Revise the introduction pointing out general diseases, not only neurodegenerative ones (lines 42-49).
- Line 62-69. It's confusing. I suppose it should be "pro-oxidants (metals) not (metal chelators)" otherwise discuss it better. Remove Ref 7 that is not appropriate. Add more references at lines 67, 68, and 69.
Results and discussion
- Reformulate the Section 2.2. Move and revise lines 104-106 in the discussion section. Differences between DPPH and ABTS radical scavenging activities can be ascribed to reaction media. Certain bioactive compounds may not be soluble in some reaction media and exert their radical scavenging activities.
- 2.3. Define the TLC acronym. Revise it moving lines 119-123 in the discussion section.
- Line 117. (365 nm, b).
- Line 129. Uniform it with line 112.
- Table 2, line 142. No bold values nor significance are listed in the table.
- For each assay put a graph summarizing the antioxidant activity with respect to extract concentrations of all analyses of Vachiella species.
- Uniform NMR with 1H NMR along with the manuscript (lines 146, 147, 194, etc.).
- Check and correct italic “and” along with the manuscript (lines 157, 246, etc.). Add conjunction “add” in listed words (lines 193, 231).
-Table 4. Move symbols explanation under the table and explain the “?” therein.
- The discussion section needs to be greatly implemented highlighting the originality of the present work and more comparisons with literature data have to be made (eg. Results from REF. 7).
- Lines 275-277. What current study??? Add Ref. At line 275.
Materials and methods
- Why did you use 50% instead of 80% likewise 4.2.3 section?
- Table 1 must be placed in the introduction section close to its first mention. Besides, put visible lines to define each table rows.
- Section 4.2.4. I don’t understand your calculation and according to what you have written the concentration range could be wrong. You take 20ul from the extract (1000ug/ml) and add to 200ul of methanol (220ul tot; 11 folds-diluted). Then, you serially dilute it (what volume???) with 110ul of methanol and add to each well 90ul of 90uM DPPH. How is possible that the extract concentration ranged from 3.9 to 500ug/ml?
- Line 362-364. Revise the sentence and change “following” with “according to”.
- Line 372. Revise extract concentrations.
- 4.2.7. Follow the same order of the results section. Line 379. Define the extract concentration.
- Lines 404-409. Revise the paragraph and parenthesis therein that is confusing. Check the sentence for Eluent B that is not more cited following its first appearance (line 406).
Reference
- Check and revise the reference list according to the guide for authors of Molecules journal.
Author Response
We would like to thank reviewer 2 for constructive criticism, which we feel assisted in improving the quality and significance of this manuscript

Round 2
Reviewer 1 Report
The manuscript improved significantly after the last review and on my opinion it deserves to be published. However, there is still an important concern about the identification of the compounds that the authors need to clarify or correct in the paper. As previously mentioned, the glycosylation position of some flavonoids was declared but no information was provided as to how was this glycosylation position determined based on MS data. For example, in Table 4 they differentiated between flavonoid glycosides at position 7 and 3 of the aglycone. It is possible to make this differentiation based on the proportion of the Y0- and [Y0-H]- ions (see Rapid Commun. Mass Spectrom. 2009; 23: 1519–1524). However, this information was not included. If the authors are not sure about the glycosylation position of those flavonoids then they should not report it and leave it as for example Quercetin O-hexoside instead of Quercetin 3-O-hexoside. Furthermore, if Rutin and Luteolin 7-O-rutioside gave the same accurate mass value and MS2 ions, then how was the differentiation of these two compounds made?
Author Response
The reply to reviewers comments is attached as PDF file

Reviewer 2 Report
The authors revised the manuscript according to the reviewer's comments but have to revise again the extract concentrations explanation.
The author's response (The stock solution (1000 ug / ml) was changed to 10 0000 ug / ml and 200 ul to 180 ul of methanol and double diluted with 100 ul. Then 100 ul of DPPH (90 uM) was added. Making the final concentration
of extracts in the first wells 500 ug / ml.) did not correspond to that included in the main text. In any case, none of these is correct and needs more clarification.
Author Response
The reply is added as attachment